# Consumption of Non-Native Bigheaded Carps by Native Blue Catfish in an Impounded Bay of the Upper Mississippi River

Tad Locher [1], Jun Wang [2,3,4], Toby Holda [5] and James Lamer [5,*]

1   Illinois Department of Natural Resources, Division of Fisheries, 700 S. 10th St., Havana, IL 62644, USA; tad.locher@illinois.gov
2   Key Laboratory of Freshwater Aquatic Genetic Resources, Ministry of Agriculture and Rural Affairs, Shanghai 201306, China; wangjun@shou.edu.cn
3   National Demonstration Center for Experimental Fisheries Science Education, Shanghai Ocean University, Shanghai 201306, China
4   Shanghai Engineering Research Center of Aquaculture, Shanghai Ocean University, Shanghai 201306, China
5   Illinois River Biological Station, Illinois Natural History Survey, Prairie Research Institute, 704 N. Schrader Ave., Havana, IL 62644, USA; holda2@illinois.edu
*   Correspondence: lamer@illinois.edu; Tel.: +1-(309)-543-6000

**Abstract:** Adult bigheaded carps *Hypophthalmichthys* spp. have never been observed in the diets of native fishes in the Mississippi River Basin. In addition, blue catfish *Ictalurus furcatus* diet preference and foraging behavior have never been studied in the presence of non-native bigheaded carps in the Mississippi River system. We examined the gut contents of adult blue catfish (567–1020 mm, $n = 65$), captured from a Mississippi River backwater using trammel nets. All items in diets were separated and enumerated, and all fish-like diet items were genetically identified to confirm species-level ID. Bigheaded carp ages were determined by sectioning hard structures (pectoral spines, post-cleithra, and vertebrae). Adult silver carp *Hypophthalmichthys molitrix* (age 3–5, mean = 3.9 years, SE = 0.2; $n = 21$) had the highest frequency of occurrence (70%) and constituted the greatest percentage by number (58%) and weight (60%) in/of blue catfish diets. Gizzard shad *Dorosoma cepedianum* ranked second by all three measures (34%, 25%, and 26%). Finally, 50% to 100% of probable age-based sizes of silver carp exceeded gape measurements of blue catfish, suggesting scavenging was the dominant means of predation. More intensive sampling efforts are required to determine the system-wide importance of bigheaded carp in blue catfish diets.

**Keywords:** bigheaded carp; silver carp; blue catfish; predation; invasive species; Mississippi River

## 1. Introduction

Invasive species are capable of altering native aquatic food webs through competitive interaction with native biota for food and resources [1]. Although negative consequences are often emphasized, non-native introductions can at times also benefit native species as an abundant alternate forage base. Magoulick and Lewis [2] determined that invasive zebra mussels *Dreissena polymorpha* served as additional forage for three native molluscivores (blue catfish *Ictalurus furcatus*, freshwater drum *Aplodinotus grunniens*, redear sunfish *Lepomis microlophus*) in the Mississippi River. Steinhart et al. [3] reported a similar finding in Lake Michigan, showing that juvenile Smallmouth Bass *Micropterus dolomieu* predation on non-native round goby *Neogobius melanostomus*, resulted in an increased rate of growth and a quicker transition into piscivory for young of the year smallmouth bass. Recently, bigheaded carps (silver carp *Hypophthalmichthys molitrix* and bighead carp *H. nobilis*) have been shown to occur in the diets of native piscivores on the Illinois River, making up to 66% frequency of occurrence and 60% diet composition by number [4].

Bigheaded carps are large-bodied planktivores introduced to the United States in the 1970s to enhance aquaculture production in the southern United States [5]. They

subsequently escaped confinement, reproduced, and increased exponentially throughout the Mississippi River Basin [6]. Since their rapid expansion, diet overlap [7] and reduced condition of native fishes [8,9] have been attributed to their introduction. Although juvenile bigheaded carps have been observed in the diets of native piscivores in the field [4] and sometimes selected for in the lab [10,11], the potential for adult bigheaded carp to be a food source for native fishes is unknown. However, blue catfish captured incidentally in Pool 26 of the upper impounded reach of the Mississippi River in 2011 were observed regurgitating adult bigheaded carps from their diets (James T. Lamer, unpublished).

Blue catfish is a large, long-lived omnivorous fish native to the Mississippi River Basin that is capable of reaching over 45 kg [12]. Due to their aggressive disposition [12] and preference for deep, main-channel habitats in large river systems, their diets can be difficult to obtain using conventional methods. Diets of blue catfish have been well-described for populations in waters of the southern and eastern United States where bigheaded carps are not found [4,13–20]. These studies have demonstrated that the blue catfish is an opportunistic omnivore whose diet includes fish, and that blue catfish consumption of fish is higher for larger individuals in backwater habitats during winter–spring. Similar results were obtained in the lower Mississippi River when co-occurring with bigheaded carp, but no incidence of bigheaded carp consumption was reported [21]. Together, these studies provide a baseline expectation for blue catfish diet composition in the Mississippi River. However, no published information exists on the possible inclusion of adult bigheaded carps in blue catfish diets.

We hypothesized that blue catfish were consuming adult silver carp to a substantial degree. Therefore, we wanted to assess if the opportunistic blue catfish utilize this abundant resource in the upper Mississippi River, and if so, at what magnitude. Our objectives were: (1) to determine bigheaded carp frequency of occurrence and mean percent composition by number and weight in blue catfish diets, (2) to estimate the age of bigheaded carp being consumed as inferred from hard structures recovered from the diets, and (3) to evaluate gape limitation of blue catfish for bigheaded carp consumed using published length-at-age and newly estimated body depth-at-length relationships.

## 2. Materials and Methods

### 2.1. Blue Catfish Collection

Blue catfish were captured in April 2013, September 2013, and monthly from April through September 2014 in Ellis Bay. Ellis Bay is a contiguous backwater at the lower end of Pool 26 of the Mississippi River (i.e., at river km 325) and was created by the impoundment formed by Lock and Dam 26 (Figure 1).

Ellis Bay has an average depth of 2 m and a maximum depth of 3.5 m. Blue catfish had been incidentally captured in Ellis Bay in 2011 (James T. Lamer, unpublished) during targeted sampling for bigheaded carps (methods detailed in Lamer et al. [22]), and the same field sampling gear was used again in this study. Trammel nets (each net: 100 m long × 2.5 m deep; inner mesh, 10 cm bar; outer tramelling, 30.5 cm bar; Miller Net Company, Inc., Memphis, TN, USA) were set in tandem, individually perpendicular to the shoreline throughout the lake, and fished while boat crews herded fish toward the nets using percussive impacts to boat and water [23]. Soak time was reduced to limit fish stress and diet evacuation. Blue catfish were removed from trammel nets and immediately placed in a 378 L poly tank that was continually refreshed with circulating water from Ellis Bay to reduce stress. Total length of each blue catfish was recorded, and for 72% of blue catfish, their gape was measured in mm using digital calipers at both the longest vertical and horizontal points.

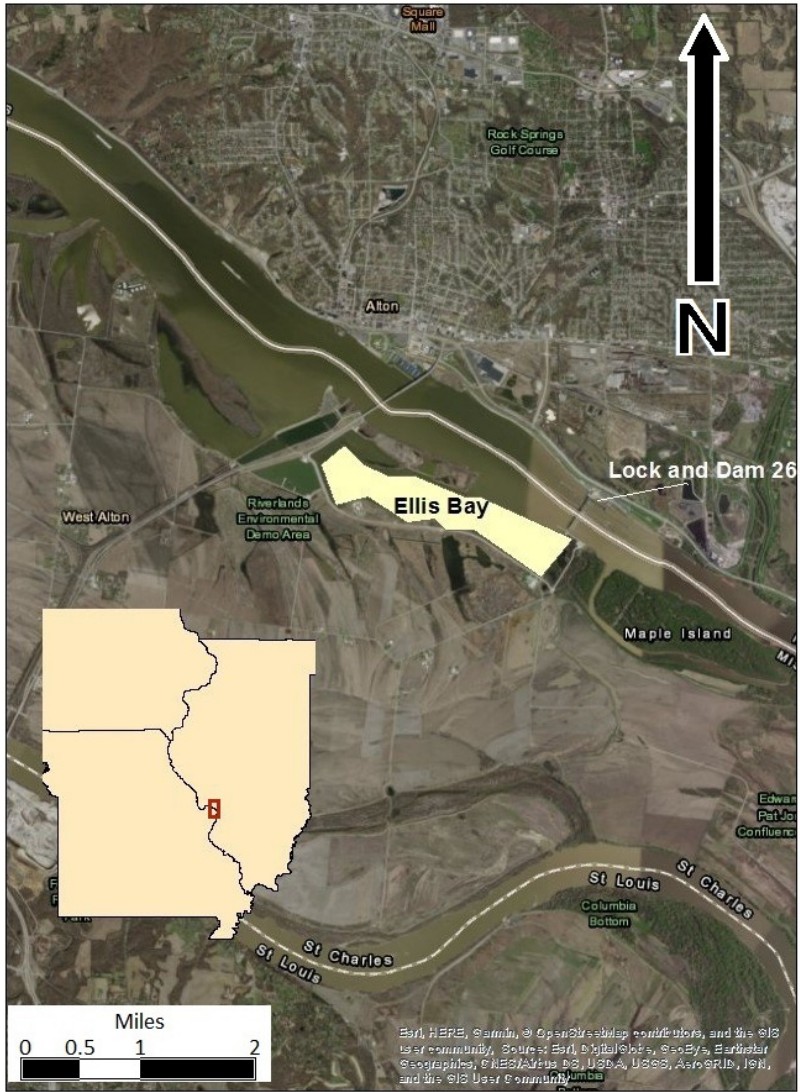

**Figure 1.** Ellis Bay study area shown in highlighted area of satellite map near Alton, Illinois, approximately 32 river km upstream of St. Louis, Missouri in Pool 26 of the Mississippi River (i.e., at river km 325). Inset shows Iowa, Illinois, and Missouri, with a red box to indicate the area of the larger satellite map shown in the figure. Note that the Missouri River is also visible in the lower portion of the satellite map.

*2.2. Diet Collection*

Gastric lavage, a minimally invasive procedure that uses pressurized water to flush contents from the stomach [24], was performed on all blue catfish using an 11 L hand-pumped compression sprayer with a modified vinyl hose attachment on the end that allowed more flexibility and easier insertion into the stomach. Diets from specimens longer than 800 mm required physical prompting by reaching into the fish's esophagus by hand and removing any large blockage that was preventing evacuation. After the blockage was removed, gastric lavage was then used to flush out the remainder of the contents. All evacuated diet contents were sealed in a 3.8 L Ziploc bag, placed on ice to retard digestion and degradation, and frozen at −10 °C within 24 h of removal. Blue catfish were released immediately after diet removal. Blue catfish with empty stomachs were recorded, and in the event of accidental mortality, whole stomachs were removed.

### 2.3. Diet Analysis

All items in the diets were separated and enumerated by the lowest identifiable taxonomic unit and dry weights obtained after drying at 110 °C to a constant weight recorded to the nearest 0.01 g [25]. No intact fishes were recovered, but large-circumference portions of silver carp were recovered, including whole silver carp heads and headless silver carp spines (caudal fin to first few vertebrae). All bigheaded carp were identified to species using any of the following structures when recovered from blue catfish diets: pharyngeal teeth, vertebrae, bones from the pelvic and pectoral girdle, presence of a ventral keel, otoliths, scales, or the presence of y-bones. Most diet items were heavily digested, making species-level identification difficult or impossible. A 5 mm × 5 mm muscle or fin biopsy was removed from all fish-like diet items for DNA analysis.

### 2.4. Genetic Analysis

DNA extraction and amplification methods were the same as those employed in Anderson et al. [4]. Fish tissue samples were placed into 96-well sample plates, and DNA was extracted using Thermo Fisher Scientific's MagMAX-96 DNA Multi-Sample Kit and following the manufacturer's protocol. Primer sequences targeted several mitochondrial domains and were based on forward and reverse versions of six published primers designed for use in fishes: (1) FishF2_t1 from Ivanova et al. [26]; (2) CytB_LC and (3) CytB_LB from Schmitt et al. [27]; (4) 12SV5-F from DeBarba et al. [28]; (5) 16S_F from Sarri et al. [29]; and (6) mlCOIintF from Leray et al. [30]. Source primer sequences were modified with added linkers as in Anderson et al. [4] (see Table S2 of that paper). Integrated DNA Technologies' PrimerQuest Tool was used to design these primers. Extracted DNA was combined with ordered primer assays in 384-well reaction plates, and mixture ratios were determined using Promega instructions for GoTaq Green Master Mix.

As in Anderson et al. [4], PCR cycles were based on the recommended protocol from each individual primer source [26–30], along with further experimentation of the primer assays on control DNA samples. A 384-well reaction plate was set up with one of the six primer assays in each plate, with reactions completed so that each sample was amplified with each of the assays. There were a total of 49 DNA samples, each amplified with the 6 assays, making a total of 294 individual PCR reactions. After initial amplification, quality control was performed by using Invitrogen 2% agarose E-gels on samples.

Products from each of the assays were then pooled together by sample, using 2 μL of product from each of the reaction plates into new 384-well plates, keeping amplifications separated by individual samples. These pooled products then underwent another PCR amplification to add unique identifying barcodes using 384 Fluidigm barcodes to distinguish between samples. Products were then pooled together a second time, with one pool per 384-well plate, so that no two samples with the same identifying Fluidigm primer sequence shared a pool. For each pool, 2 μL from each barcoded reaction were added. Sequencing was performed using a MiSeq V2 Nano, 2 × 250 sequencing run. Sequencing results for diet item taxonomic identifications were compared with visual-derived identifications when present to confirm genetic results. Items that had not been able to be visually identified were identified using sequencing results. Non-fish highly improbable assignments due to geography or taxa were discarded. The final diet item list was used in all subsequent statistical analyses.

### 2.5. Aging of Hard Structures

Intact hard structures (post-cleithra, pectoral spines, and vertebrae) recovered from bigheaded carp in the diets were dried and then sectioned (0.5 mm sections) using an Isomet low-speed saw [31]. Age was determined by the consensus annuli count for one or all structures depending on the presence of structures available in individual diets after visual inspection under a dissecting microscope. Three observers independently aged all structures and a consensus age was reached. No known-age structures or adult bigheaded

carp validation exist in the literature. However, the three independent observers were experienced in aging invasive carp structures.

### 2.6. Data Analysis

Blue catfish that were captured and found to have empty stomachs were excluded from diet analyses. Species identification of all prey items was confirmed with genetic analyses and associated with measured dry weight of diet items. Frequency of occurrence was determined for all diet items and was defined as the percentage of blue catfish stomachs that contained a particular prey item, genetically identified to the lowest taxonomic rank, irrespective of the amount [25]. In addition to frequency of occurrence, we estimated the mean percent composition by number, primarily for comparison to values in Anderson et al. [4], and due to potential digestion-related bias in the following percent composition by weight. We also estimated the mean percent composition by weight. Mean percent composition by weight refers to the percentage of the total dry weight that each individual diet component, identified to the lowest taxonomic rank, comprises. Note that this metric may be biased toward less digestible prey in our study due to the highly digested nature of recovered gut contents.

### 2.7. Size of Prey and Gape Limitation

We wanted to compare the sizes of silver carp to gapes of blue catfish to determine the possible extent of the whole predation. The appropriate measures for comparison are the widest point of a silver carp (body depth) and the larger gape of a blue catfish (horizontal gape). We could not obtain measurements of body depths of consumed silver carp; however, we did obtain age estimates for 21 silver carp, which can be used to approximate total length via established length-at-age relationships [32]. Williamson and Garvey [32] published a length-at-age relationship for silver carp based on observations in the middle Mississippi River during 1998–2001. However, higher density populations may result in smaller lengths at age [33], and Ellis Bay in 2013–2014 likely had higher densities of silver carp than observed in the middle Mississippi River in 1998–2001 [32]. Therefore, we also used the length-at-age values published for the Amur River in Russia using data originally published by Nikoslkii [32,34], because this length-at-age curve gave smaller lengths-at-age than that for the middle Mississippi River, and it seems unlikely that silver carp would ever show a lower length-at-age than in the Amur River. Therefore, the Amur River l length-at-age is likely to provide the lowest possible percent of silver carp that were gape limited to their consuming blue catfish in our study.

To compare silver carp sizes with blue catfish gape measurements, we needed a conversion from silver carp total length to silver carp body depth. We took morphometric measurements of 95 silver carp collected from the LaGrange and Peoria reaches of the Illinois River during October 2021 (see Table S3). Using these data, we fit a linear regression of silver carp maximum body depth as a function of total length ($r^2 = 0.79$, body depth = 0.2 total length + 17.1). We then predicted silver carp body depth from approximate lengths obtained for each silver carp consumed by a blue catfish in Ellis Bay.

We had horizontal gape estimates for 12 of the 21 blue catfish for which we had age estimates of consumed silver carp. Thus, we had to predict the expected horizontal gape for the remaining 9 blue catfish using the total length measurements we did have. We developed a prediction equation by fitting a linear regression to horizontal gape and total length data for the 47 blue catfish for which we had both measurements ($r^2 = 0.87$, horizontal gape = 0.18 total length − 41.7). Finally, we compared our estimates of approximate body depths of individual age-estimated silver carp to the measured or predicted horizontal gapes of each respective individual blue catfish. We report the percent of silver carp that exceeded their consumer's gape limitation both on the basis of the middle Mississippi River length-at-age curve and also on the basis of the Amur River length-at-age curve.

## 3. Results

### 3.1. Blue Catfish Collections and Diet Identifications

Blue catfish were captured at every sampling period in Ellis Bay in 2013 (*n* = 32) and 2014 (*n* = 33) and varied in total length from 567 mm to 1020 mm and in wet weight from 2.15 kg to 21.23 kg (Table S1). Vertical and horizontal gapes were measured on 72% (47 out of 65) of blue catfish and ranged from 57 mm to 150 mm (horizontal) and 47 mm to 115 mm (vertical) (Table S1). Of the 65 fish captured, 12 (19%) had empty stomachs. Of the 53 blue catfish with quantifiable diets, 44 (83%) were captured in April 2013 and April 2014 (23 and 21, respectively).

A total of 50 diet items were able to be conclusively identified apart from further genetics analysis (Table 1). The remaining 48 (including 1 visually unidentifiable sample and 47 samples preliminarily identified as silver carp) were subjected to genetic analysis, from which most turned out to be silver carp and gizzard shad, but instances of freshwater drum, bigmouth buffalo, bighead carp, and bluegill were also observed (Table 1).

**Table 1.** Diet item identification using visual and genetics methods, with count of items in each row and count of diet items with aged hard structures. Items not submitted for genetic analysis (i.e., confident visual identification) have an "NA" in the Genetics ID row. Note that age estimates for the three genetically identified gizzard shad included two based on vertebrae and one based on a pectoral spine (i.e., none based on cleithra, which are characteristic of *Hypophthalmichthys* spp.).

| Visual ID | Genetics ID | N | N Aged |
|---|---|---|---|
| Silver carp | NA | 27 | 17 |
| Gizzard shad | NA | 12 | 0 |
| Freshwater drum | NA | 6 | 0 |
| Crawfish | NA | 3 | 0 |
| Bighead carp | NA | 1 | 1 |
| Coot | NA | 1 | 0 |
| Silver carp | Silver carp | 12 | 4 |
| Silver carp | Gizzard shad | 22 | 3 |
| Silver carp | Freshwater drum | 3 | 0 |
| Silver carp | Bigmouth buffalo | 2 | 0 |
| Silver carp | Bighead carp | 1 | 0 |
| Silver carp | Bluegill | 1 | 0 |
| Unknown | Bluegill | 1 | 0 |

### 3.2. Diet Proportion, Age, and Gape Limitation Analyses

Bigheaded carps had the highest frequency of occurrence in the diets (73.6%, *n* = 39), with silver carp comprising 69.8% (*n* = 37) and bighead carp comprising 3.8% (*n* = 2) (Figure 2). Bigheaded carps were also the dominant diet component in percent composition by number (Figure 3) and by dry weight (Figure 4), as silver carp accounted for 58.4% and 59.7%, and bighead carp accounted for 1.4% and 2.7%. Gizzard shad *Dorosoma cepedianum* were the second most common diet component, occurring in 34% (*n* = 18) of blue catfish diets and making up 25% and 26.3% of the percent diet composition by number and by weight.

A total of 27 structures (post-cleithra = 6, vertebrae = 11, and pectoral spines = 9) from 21 individual silver carp were found and aged, as well as 1 post-cleithra from a bighead carp and 2 vertebrae and 1 pectoral spine from 3 gizzard shad (Table S2). Observed silver carp ages reached by three readers ranged from 3 to 5 years with a mean consensus age of 3.9 (SE = 0.2) years (Table S2). When silver carp total length approximation was based on middle Mississippi River length-at-age values, 100% (21 of 21) exceeded their consuming blue catfish's horizontal gape (Figure 5A). When silver carp total length approximation was based on Amur River length-at-age values, 52% (11 of 21) would have exceeded their consuming blue catfish's horizontal gape (Figure 5B).

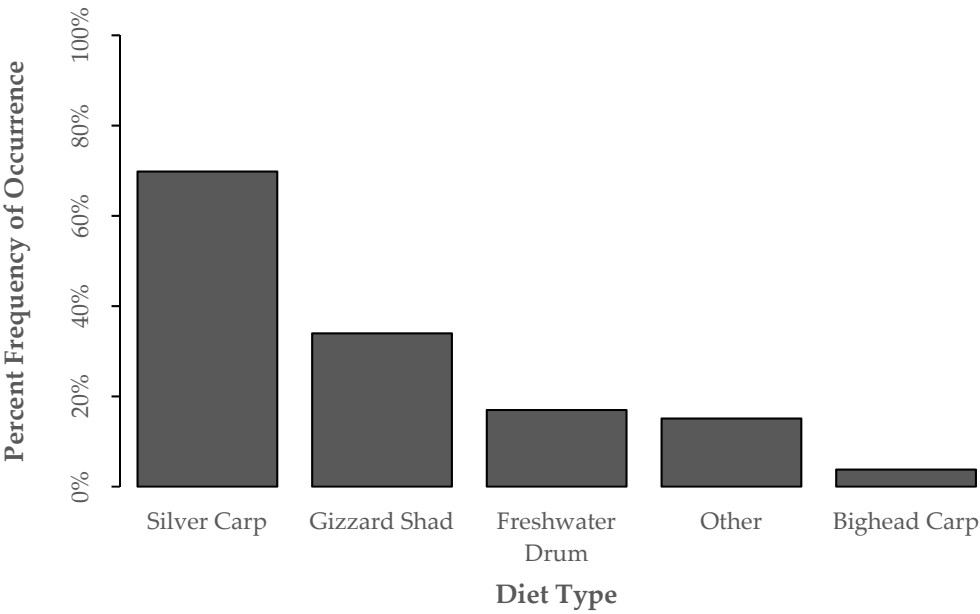

**Figure 2.** Percent frequency of occurrence of diet items in blue catfish (*Ictalurus furcatus*) with non-empty stomachs (*n* = 53) sampled from Pool 26 of the Mississippi River near Alton, Illinois, starting in April 2013 and continuing through October 2014. Silver carp (*Hypophthalmichthys molitrix*) (69.8%), gizzard shad (*Dorosoma cepedianum*) (34.0%), freshwater drum (*Aplodinotus grunniens*) (17.0%), other (15.1%), and bighead carp (*Hypophthalmichthys nobilis*) (3.8%). The 'other' category includes all diet items (fish and non-fish) with <10% frequency of occurrence with the exception of bighead carp, which were a focus of the study.

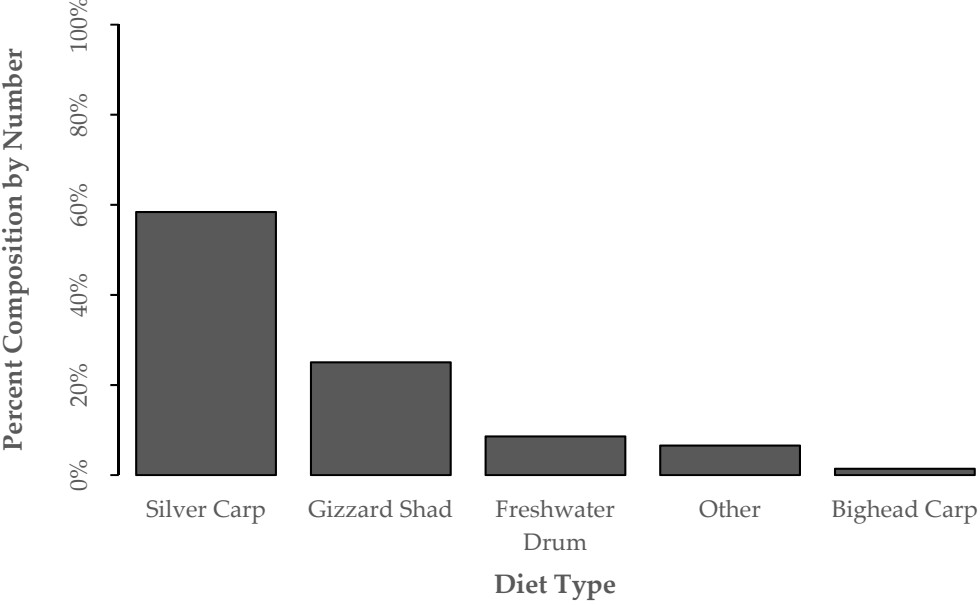

**Figure 3.** Percent composition by number of diet items in blue catfish with non-empty stomachs (*n* = 53) sampled from Pool 26 of the Mississippi River near Alton, Illinois, starting in April 2013 and continuing through October 2014. Silver carp (58.4%), gizzard shad (25.0%), freshwater drum (8.6%), other (6.6%), and bighead carp (1.4%). The 'other' category includes all diet items (fish and non-fish) with <5% composition by number with the exception of bighead carp, which were a focus of the study.

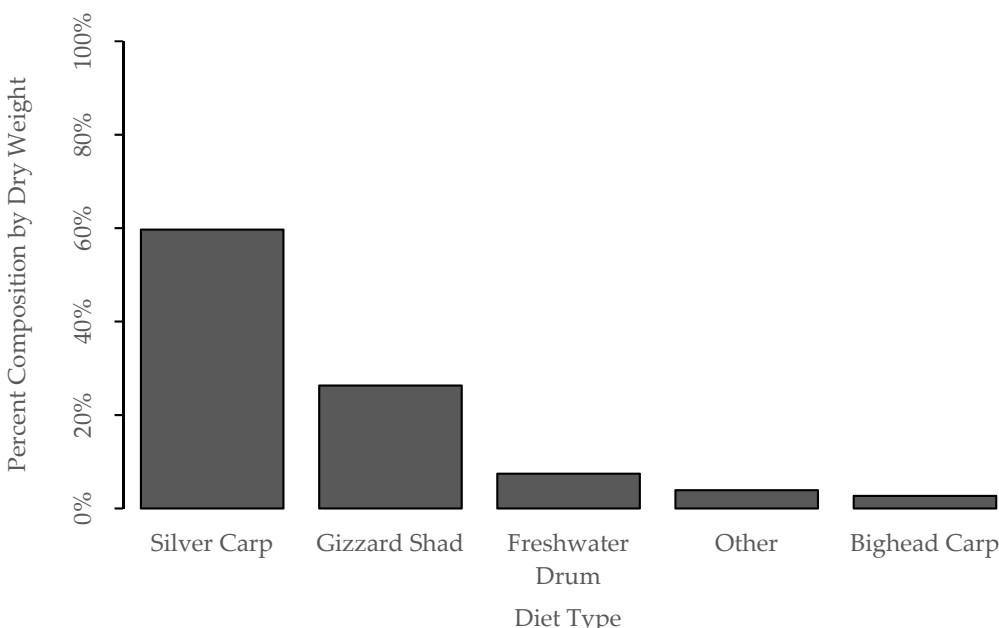

**Figure 4.** Mean percentage of diet components by dry weight (g) collected from blue catfish in Pool 26 of the Mississippi River near Alton, IL, from April 2013 through October 2014. Silver carp (59.7%), gizzard shad (26.3%), freshwater drum (7.4%), other (3.9%), and bighead carp (2.7%). The 'other' category includes all diet items (fish and non-fish) with <2.5% composition by dry weight.

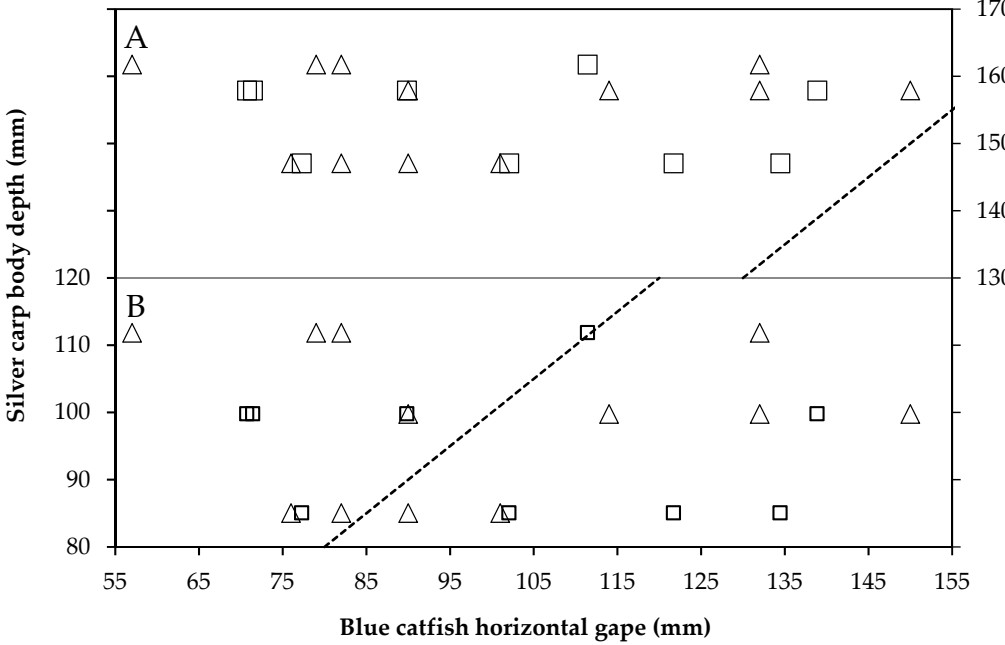

**Figure 5.** Age-predicted silver carp body depth plotted against measured (triangle points) and estimated (square points) horizontal gape of the consuming blue catfish for the 21 blue catfish, which contained age-able silver carp in their stomachs. The difference in panels (**A**,**B**) are in which length-at-age curve was used to convert silver carp age estimates to silver carp body depth estimates. In panel (**A**), length at age was based on middle Mississippi River data collected during 1998–2001 [32]. In panel (**B**), length at age was based on data collected prior to 1954 from the Amur River in Russia [32,34].

## 4. Discussion

This study is the first to document the inclusion of adult/sub-adult (3- to 5-year-old) invasive bigheaded carp in the diets of any native fish. Previous documentation of invasive bigheaded carps consumed by native fishes were all for YOY bigheaded carps < 100 mm TL [4,10,11]. The current study also reports a rate of occurrence of invasive bigheaded carps in native fish diets (70%) exceeding even the highest estimate in the only other field study reporting such estimates (65% for white bass *Morone chrysops* in Aug–Sep reported in [4]). Percent composition by number of bigheaded carps in blue catfish diets (60%) was comparable to values for white bass (61%) and shortnose gar *Lepisosteus platostomus* (48%) consuming YOY as reported by Anderson et al. [4], but substantially higher than for other fishes in that study. This is also the first study to document the inclusion of invasive bigheaded carp of any age in the diets of blue catfish. Further, we report that in blue catfish diets, the rates of frequency of occurrence and contribution by weight for bigheaded carp were the highest of any taxa, exceeding the same values for gizzard shad over two times.

The greater dependence of blue catfish on bigheaded carps than on gizzard shad in Ellis Bay can likely be attributed to the high abundance of bigheaded carps. As opportunistic omnivores, blue catfish would be expected to forage heavily on any suitable food resources abundant enough to be encountered frequently [12–14,21]. Pool-wide electrofishing CPUE for silver carp in Pool 26, which contains Ellis Bay, was among the highest on record in 2014 and was still moderately high in 2013, similar to the pattern observable in the un-impounded reach just downriver [35,36]. The high abundance of this prey source likely results in more frequent encounters. Thus, based on silver carp abundance alone, and given the obvious fact that blue catfish can consume them, it is unsurprising that blue catfish were consuming silver carp so heavily in Ellis Bay in Pool 26 of the Mississippi River during 2013–2014.

One of the most startling results of this study is the occurrence of adult-age bigheaded carps in the diets of blue catfish. Large chunks and some nearly intact silver carp were recovered from blue catfish diets, but it was unclear in the field whether silver carp were being consumed whole. Our gape limitation analysis indicates that none of the silver carp (or at best a minority of 46%) were small enough for whole live consumption to be the probable means of consumption by their respective blue catfish. If blue catfish were not consuming whole live bigheaded carps, the most likely conclusion is that blue catfish were scavenging injured, dying, or dead silver carp that were already cut up or else easily crushed. Graham [12] reported two communications regarding blue catfish being known to scavenge dead, dying, and injured gizzard shad. Eggleton and Schramm [21] included scavenging as a category of diet composition. However, it seems strange that so many adult bigheaded carps would be regularly available for scavenging. It would be interesting to test whether blue catfish are able to exceed their gape limitation, perhaps by crushing prey, and might therefore be able to actively pursue and prey on adult bigheaded carps. If so, managing abundant blue catfish populations could contribute to management efforts seeking to limit the spread and reduce abundances of invasive bigheaded carp populations.

While our study may indicate blue catfish were scavenging when consuming bigheaded carps, it would mean a higher degree of reliance on scavenging (61% by dry weight) than the 6% scavenging reported for blue catfish by Eggleton and Schramm [21]. Why should blue catfish in our study be scavenging at so much higher a rate? If bigheaded carp are abundantly available to be scavenged, their high availability might explain an opportunistic predator's high reliance. However, it may also make sense for blue catfish to scavenge any available silver carp from an energetic standpoint. For example, if bigheaded carp and gizzard shad were both available to be scavenged, it would likely take blue catfish a similarly low energy output to acquire and ingest them. However, energy return would be higher for a bigheaded carp (or even part of one) than for a gizzard shad. The energy density of silver carp (5200 J/g [37]) is similar to, if slightly lower than, that of gizzard shad (5476 J/g; [38]). However, the mass of an average 3-year-old silver carp (about 3000 g wet weight for a 650 mm TL 3-year-old [32,39]) is about six times much greater than the

mass of the largest gizzard shad specimens (450 g wet weight [40]). Thus, the energetics of scavenging opportunities might encourage blue catfish to rely more heavily on bigheaded carp. However, energetics has been demonstrated to be relatively ignored compared to mere availability in the foraging decisions of blue catfish [21]. Whether for energetics or merely due to availability, high scavenging reliance of blue catfish on bigheaded carp would indicate that injured/dying/dead bigheaded carp were highly available.

Previous studies on predation of juvenile bigheaded carps by native piscivores only show a temporary utilization of the forage base after a significant spawning event, and then those fish were quickly able to outgrow predation within the first or second year [4,11]. Because the magnitude of young of year bigheaded carp availability is highly variable, bigheaded carps cannot be relied on as a consistent forage base for these smaller predators. Blue catfish can consume adult bigheaded carp, and they may be able to consume them year-round and even in years of poor year-class strength. We are not able to test this, as most of our blue catfish were collected in spring. It would therefore be useful to expand this study temporally.

In this study, blue catfish were collected exclusively from a backwater habitat. The importance of these backwater habitats has been documented on multiple occasions, showing higher species richness [41], higher energy content of blue catfish prey items [22], and increased blue catfish growth rates [42]. However, the backwater scope of the current study limits our ability to make general statements about blue catfish consumption of bigheaded carps in the main channel of the upper Mississippi River. Clearly, it would be useful to expand this study with diets from the main channel from more sites.

**Supplementary Materials:** The following supporting information can be downloaded at: https://www.mdpi.com/article/10.3390/fishes7020080/s1, Table S1: Blue Catfish Data; Table S2: Diet Items Data; Table S3: Silver Carp Morphology Data.

**Author Contributions:** Conceptualization, T.L. and J.L.; methodology, T.L., J.L. and T.H.; formal analysis, T.L. and T.H.; investigation, T.L and T.H.; resources, J.L. and J.W.; data curation, T.L. and T.H.; writing—original draft preparation, T.L.; writing—review and editing, T.H., J.L. and T.L.; visualization, T.L. and T.H.; supervision, J.L.; project administration, J.L. and T.L.; funding acquisition, J.L. All authors have read and agreed to the published version of the manuscript.

**Funding:** This research received no external funding.

**Institutional Review Board Statement:** The animal study protocol was approved by the Institutional Animal Care and Use Committee of Western Illinois University (Protocol Code: 16-10r; date of approval: 24 August 2016).

**Data Availability Statement:** The data presented in this study are available in Tables S1–S3.

**Acknowledgments:** We thank the members of the fisheries laboratory at Western Illinois University, volunteers from Western Illinois University's Kibbe Field Station, and technicians from the Illinois Natural History Survey, National Great Rivers Research and Education Center. Without their help with the sampling efforts, none of this research would have been possible.

**Conflicts of Interest:** The authors declare no conflict of interest. The funders had no role in the design of the study; in the collection, analyses, or interpretation of data; in the writing of the manuscript, or in the decision to publish the results.

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
