# Peer review of "Consumption of Non-Native Bigheaded Carps by Native Blue Catfish in an Impounded Bay of the Upper Mississippi River"

_fishes, doi:10.3390/fishes7020080_

Round 1

Reviewer 1 Report

Lines 33-35. “Although negative consequences are often emphasized, nonnative introductions can also provide some positive benefits as an abundant alternate forage base for native fishes.” I am sorry, but I disagree with the authors. More potential food for a given species cannot be considered a benefit in ecological and evolutionary terms. Such thinking may be possible for a fish farmer or breeder in zootechnical terms, but from an ecological and evolutionary point of view, no benefits can be claimed. The introduction of a species is an alteration of the natural ecological and evolutionary succession, and the fact that it serves as food for a particular species cannot be used as an argument for its justification. The authors should either eliminate such reasoning or explain that it is not acceptable in ecological and evolutionary terms.

Lines 69-75. The aims of the manuscript are purely descriptive and there are no hypotheses. The authors should try to present the objectives of their research by stating a certain number of hypotheses and the means to test them.

Lines 120-152. Neither the manuscript nor the supplementary material includes any results of the genetic analysis. We can only believe that these analyses have been done, but any methodology should be accompanied by some results.

Lines 162-175. Statistical analysis does not exist as such. Only percentages are included in the Results. Perhaps this section should more properly be called "Data analysis", "Numerical data" or something similar, but including the term statistics seems inappropriate.

Line 238. “…(g) (± 2 SE) collected…”. The figure shows only mean values and no standard error values. "(± 2 SE)" should be deleted.

Lines 248-250. I think that Table 1 is not very relevant to the understanding of the manuscript. I find it somewhat confusing, and I do not understand how the data have been arranged. It seems to me to be raw data. I think it could be removed from the text and included in the supplementary material.

Author Response

Thank you for taking time to review our manuscript.

Please see the attachment for our responses to your comments.

Reviewer 2 Report

Dear authors,

these are the comments on your submitted paper. 

Abstract

Line 14  Change ssp. to ssp. and in the same sentence please add some temporal determinant. Some readers may conclude from your opening sentence that you didn’t find any bigheaded carps in your study.

Line 36  Please put the Latin names of species in italics and provide an author of the species on the first mention. Please do this throughout the manuscript.

Materials and methods

Please provide a higher resolution figure for the map which includes scale bar and indicates North

Line 88  How many trammel nets? All together 100 m or the individual net was 100 m long?

Where the fish anesthetized during the procedure?

Line 95  Please provide more detail how the gape was measured and in which units.

Line 96  How many Blue Catfish in total were caught and measured and why didn’t you measure the horizontal and vertical gape for all the individuals?

Last paragraph of materials and methods is very confusing. I don’t understand why you had to estimate horizontal gape for fish that you have caught? Was it measurement or estimate? In line 96 you mentioned that you measured both horizontal and vertical gape for 76% of all caught Blue Catfish and now you mention that you estimated 12 of 21 (over 50%). Is 21 number of  Blue Catfish with Bigheaded Caps in their stomachs? And if you estimated 12 of 21 how can you predict the remaining 10? That would be 22 in total.

Results

Line 210               is the length total or standard?

Line 209-215      in 2013 you caught 32 specimens and in 2014, 33. How is total 62 (47 out of 62)?

Figures 2., 3. i 4. Please provide abbreviation meaning in the figure caption

Figure 5a is completely missing from the results section.

Discussion

Line 265 Please provide a Latin name and author for White Bass

Author Response

(The authors gave the same response as above.)

Reviewer 3 Report

The paper is well written and I can only find a few minor things to be edited – in whole Introduction section all scientific Latin names should be in italics. In line 212 you missed a space before 57 mm. In figures 2-4 you use abbreviations on a chart for common species names, but although intuitive, there is no explanation of their meaning in figure captions.

Author Response

(The authors gave the same response as above.)

Round 2

Reviewer 2 Report

Dear authors, you have successfully addressed all my concerns and MS is now clearer and easier to follow.

Just a few more smaller concerns:

The is still no metric unit in lines 106 and 107 of the revised MS.

Are the fonts of panel A and B deliberately of different sizes on Figure 5.?

Please rephrase the caption for Fig. 5.  It is too lengthy and from the first sentence I would conclude that you could predict the body depth of silver carps using the mouth gape of blue catfish which is surely not the case.

What is the point of panel B when there is no blue catfish in Russia and same species in different conditions grows at different rates. If you want to keep the panel B then please discuss the point in the discussion section.   

Author Response

Thank you again for your review.

Please see the attachment for our responses.
